# A Comprehensive Evaluation of Large Language Models on Legal Judgment Prediction

**Ruihao Shui**
National University of Singapore
ruihaoshui@u.nus.edu

**Yixin Cao**
Singapore Management University
yxcao@smu.edu.sg

**Wang Xiang**[*]
University of Science and Technology of China
xiangwang1223@gmail.com

**Tat-Seng Chua**
National University of Singapore
dcscts@nus.edu.sg

## Abstract

Large language models (LLMs) have demonstrated great potential for domain-specific applications, such as the law domain. However, recent disputes over GPT-4's law evaluation raise questions concerning their performance in real-world legal tasks. To systematically investigate their competency in the law, we design practical baseline solutions based on LLMs and test on the task of legal judgment prediction. In our solutions, LLMs can work alone to answer open questions or coordinate with an information retrieval (IR) system to learn from similar cases or solve simplified multi-choice questions. We show that similar cases and multi-choice options, namely label candidates, included in prompts can help LLMs recall domain knowledge that is critical for expertise legal reasoning. We additionally present an intriguing paradox wherein an IR system surpasses the performance of LLM+IR due to limited gains acquired by weaker LLMs from powerful IR systems. In such cases, the role of LLMs becomes redundant. Our evaluation pipeline can be easily extended into other tasks to facilitate evaluations in other domains. Code is available at https://github.com/srhthu/LM-CompEval-Legal

## 1 Introduction

Large language models have achieved great success in various Natural Language Processing (NLP) tasks (Brown et al., 2020; Touvron et al., 2023), while there are still some disputes over the potential for domain-specific applications (Martínez, 2023). Focusing on the law domain, the leading LLM, GPT-4 (OpenAI, 2023), was claimed to pass the Uniform Bar Exam (UBE) with a 90th percentile score. Although inspiring, however, this result was pointed out to be overestimated (Martínez, 2023).

---
[*]Xiang Wang is also affiliated with Institute of Artificial Intelligence, Institute of Dataspace, Hefei Comprehensive National Science Center.

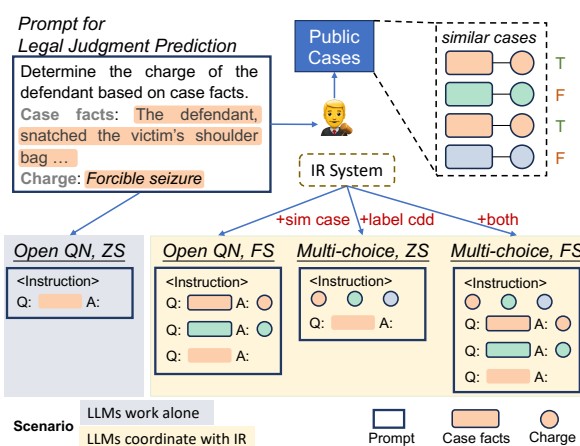

Figure 1: The task of Legal Judgment Prediction and the evaluation settings. Different colors refer to different charges. For similar cases, "T" refers to true similar cases with the same charges as the query cases, while "F" refers to false similar cases. For task settings, "ZS" is the abbreviation for zero-shot and "FS" for few-shot.

This raises an interesting question: How exactly LLMs perform in various real-world legal tasks?

In this paper, we design practical baseline solutions based on LLMs and systematically investigate their competency in the law, to shed light on other domains as well. We attribute the main issues of the previous benchmark as follows. First, UBE is too general and not subject to any legal jurisdiction (Martínez, 2023). Second, UBE contains multi-choice questions and open-ended questions that require human experts to evaluate. To avoid human evaluation, some datasets (Hendrycks et al., 2020) replace open-ended questions with multi-choice questions. However, in real-world applications, there are not only multi-choice but also open questions. Using multi-choice questions only may not be comprehensive enough. Third, specifically in but not limited to common law (Shulayeva et al., 2017; Xiao et al., 2019), similar cases are always introduced as evidence to support expertise legal reasoning (Zhong et al., 2020b), which are not fully

studied in previous benchmark (Hendrycks et al., 2020).

For the first issue, we choose legal judgment prediction (LJP) (Xiao et al., 2018; Chalkidis et al., 2019; Zhong et al., 2020a) as the example task for investigation. It is a real-world problem to determine the charges committed by the defendants under a juridical system, as shown in Figure 1. LJP is typically formulated as a classification task to predict the most possible one from a list of pre-defined charges. Then, for the second and third issues, we design four settings derived from two work scenarios of LLMs to cover open and multi-choice questions and the usage of similar cases. In the first scenario, LLMs **work alone** without explicit knowledge in prompts, assuming all domain knowledge is implicitly stored in parameters. In the second scenario, LLMs **coordinate with an information retrieval (IR) system** that enriches prompts with similar demonstrations and label candidates to benefit expertise reasoning. Specifically, demonstrations consist of pairs of similar cases and their charges, which are retrieved by the IR system based on similarity of case facts. Labels of the retrieved cases can form *label candidates*, shown as circles of different colors in Figure 1, to hint LLM with label information and narrow down label space (Ma et al., 2023).

The four evaluation settings in Figure 1 can be categorized based on the presence of two elements in prompts: demonstrations (similar cases) and label candidates. Demonstrations convert the setting from zero-shot to few-shot prompting, while label candidates simplify the task from open questions to multi-choice questions[1]. The first scenario corresponds to the first setting, where neither element is present, while the second scenario encompasses the remaining three settings. We evaluate five up-to-date LLMs of the close-source GPT-3 (Brown et al., 2020) family, ChatGPT and GPT-4 (OpenAI, 2023), and open-source LLMs including Vicuna (Chiang et al., 2023), ChatGLM (Du et al., 2022) and BLOOMZ (Muennighoff et al., 2022). The evaluation is conducted on a Chinese LJP dataset, namely CAIL (Xiao et al., 2018), which contains cases of 112 criminal law charges[2].

We highlight our key findings as follows:
1. Similar cases and label candidates can help

LLMs recall domain knowledge that is critical for expertise legal reasoning.
2. Label candidates result in more consistent outputs, indicating LLMs gain greater confidence in their domain knowledge (Jiang et al., 2021).
3. Irrelevant demonstrations formed by fixed cases hardly improve performance. This excludes their effect on task illustration.
4. Paradox: An IR system can outperform LLM+IR since weaker LLMs acquire limited gains from informative documents retrieved by a powerful IR system. Thus, it is critical to adapte LLMs to generate with retrieved documents.
5. More similar cases introduce more knowledge and noise simultaneously, whose final outcome depends on LLMs.

The main contributions are summarized in three aspects:

- We investigate the law competency of LLMs on the task of legal judgment prediction.

- We propose practical baseline solutions for LLMs that tackle two scenarios: working alone or in coordination with an IR system.

- We evaluate five LLMs and conduct comprehensive analysis to demystify their characteristics of expertise reasoning.

## 2 Baseline Method

The goal of legal judgment prediction is to determine the committed charges given case facts. To harness LLMs for LJP, we adopt in-context learning (Brown et al., 2020) and use LLMs to generate the charges conditioned on prompts (Section 2.1). To enhance LLMs, we incorporate label candidates and demonstrations consisting of similar cases into prompts, which are acquired by an IR system (Section 2.2). This derives **four settings** of baseline solutions, namely zero-shot open questions, few-shot open questions, zero-shot multi-choice questions, and few-shot multi-choice questions. The multi-choice settings employ label candidates while few-shot settings include demonstrations, as shown in Figure 1. Finally, we introduce how to simulate IR systems with different capabilities to understand their effects (Section 2.3).

### 2.1 LLM Prompting

**Prompt Design.** A prompt begins with an instruction to illustrate the task followed by label

---

[1] It is not strict multi-choice questions. LLMs can generate correct answers even though ground-truth labels are absent in candidates.

[2] After filtering less frequent (article, charge) pairs

candidates and task demonstrations in the form of input-output pairs. The templates of prompts are displayed in Appendix A.1.

**Parsing.** We adopt one automatic parsing function for all LLMs to map LLM outputs to pre-defined charge labels. No ad hoc heuristics are employed for a fair comparison. Specifically, we use the BM25 algorithm[3] to measure text similarity between outputs and pre-defined charges and predict the most similar charges. BM25 is robust and yields comparable performances to neural similarity methods like `text2vec`[4] in our pilot experiments.

**Inference.** Sampling is enabled during generation for consistent results, as inspired by Wang et al. (2022). Five outputs are sampled for each prompt with the temperature of $0.8$. Their similarity scores of pre-defined labels are averaged.

## 2.2 IR System for Knowledge Incorporation

IR systems are utilized to retrieve *similar cases*, commonly referenced by lawyers and judges, to inform their judgments. In addition to providing demonstrations, these similar cases can also aid in generating potential labels by incorporating the labels from the top similar cases. By employing these smaller sets of predefined charges, namely *label candidates*, complex open questions can be simplified into multiple-choice questions. This approach is effective in enhancing LM prompting (Ma et al., 2023), as including hundreds of charges directly in prompts is impractical.

**Implementation of IR System.** We use the BM25 algorithm to measure the semantic similarity between cases. Similar cases are retrieved from the training dataset. To guarantee that the demonstrations exemplify one of the multi-choice options, we exclude demonstrations with labels that are not among the candidate options[5].

## 2.3 Simulation of IR Systems

To investigate the effects of IR capabilities, we simulate a series of IR systems of different capabilities as measured by Precision@1[6]. Then the top retrieved cases are used as demonstrations. We consider cases with identical charges to the query cases as true similar cases and vice versa.

**Realistic Simulation.** We prioritize the returning of *true* similar cases for *easy* query cases, rather than the returning in a random manner. The query difficulty is measured by the Precision@10 of the BM25 retriever described in Section 2.2. The motivation is that queries with shadow linguistic features are more possible to get relevant retrieval results than complex or obscure queries. For a specific value (*e.g.,* a%) of Precision@1 to be simulated, the top a% of easy test cases are assured to have a *true* similar case, while the rest are assigned *false* similar cases.

# 3 Experimental Setup

## 3.1 Models

Below is a concise introduction to the five LLMs to be evaluated.

**GPT-4** (OpenAI, 2023) and **ChatGPT** are available from OpenAI API and the versions of `gpt-4-0314` and `gpt-3.5-turbo-0301` are used. For technological details, ChatGPT is claimed to be a sibling model to InstructGPT (Ouyang et al., 2022) that is trained to follow instructions and align to human preferences with the RLHF algorithm (Christiano et al., 2017).

**Vicuna-13B** (Chiang et al., 2023) is a LLaMA model (Touvron et al., 2023) fine-tuned on 70K public user-shared conversations with ChatGPT. It can be viewed to learn distilled knowledge (Hinton et al., 2015) of ChatGPT.

**ChatGLM-6B**[7] is a dialog language model based on the GLM (Du et al., 2022) architecture and supports English and Chinese.

**BLOOMZ** (Muennighoff et al., 2022) is an instruction fine-tuned BLOOM (Scao et al., 2022), a multilingual language model. We use the `bloomz-7b1-mt` version that is tuned for multilingual prompts. Except for BLOOMZ, Vicuna and ChatGLM are mainly fine-tuned on conversational data.

## 3.2 Dataset and Pre-processing

The Chinese LJP dataset, CAIL (Xiao et al., 2018), is used in our experiments. Each sample consists of the case *facts* and the committed *charge* as the label. As the original dataset is very large (~100K for training and ~20K for test), we randomly sample a balanced small test set from the original test set. Five cases are sampled for each charge, accounting

---

[3] https://pypi.org/project/rank-bm25/

[4] https://github.com/crownpku/text2vec

[5] This condition is not violated for the top four similar cases without filtering.

[6] The accuracy of the top one retrieved case.

[7] https://github.com/THUDM/ChatGLM-6B

| Tokenizer | Median | <=500 | <=1000 |
|-----------|--------|-------|--------|
| ChatGPT | 396.5 | 68.75 | 92.32 |
| Vicuna | 496.0 | 50.89 | 86.96 |
| ChatGLM | 206.5 | 91.07 | 98.57 |
| BLOOMZ | 210.5 | 90.54 | 98.93 |

Table 1: Statistics of the number of tokens across tokenizers. The last two columns present the ratios of test samples with token counts below the specified values.

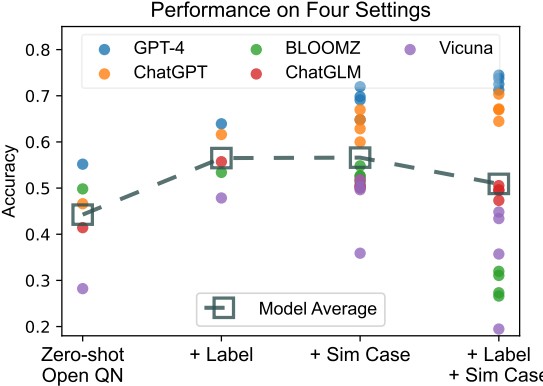

Figure 2: The macro comparison between the four settings. "+Label" refers to zero-shot multi-choice questions; "+Sim Case" refers to few-shot open questions and "+Label +Sim Case" refers to few-shot multi-choice questions. More than one points of a model in the last two settings refer to runs with different number of demonstrations.

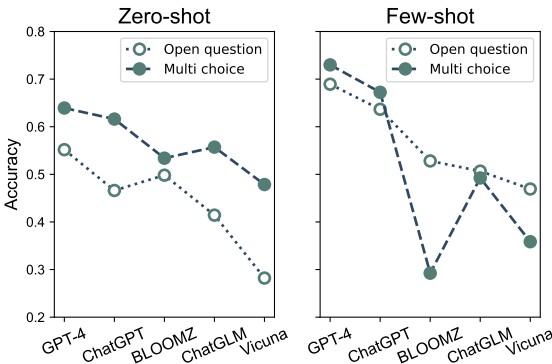

Figure 3: Compare the models under each setting. Few-shot performances are averaged among 1-shot to 4-shot.

for 560 test cases in total for 112 charges. Similarly, we also sample the training and validation sets with 10 cases per charge. The training set is used to retrieve similar cases (Section 2.3), while the validation set is used to determine the optimal $k$ of the $k$NN algorithm.

**Truncation.** Since some cases have very long descriptions, we truncate the case facts of demonstrations to 500 tokens and those of test samples to 1000 tokens. It is worth noting that the text is tokenized by the tokenizer of each model before truncation for a fair comparison. Recently, Petrov et al. (2023) address the issue that a tokenizer can lead to different performances of different languages. This suggests that the performance on a particular language can also be influenced by tokenizers from various models with varying language encoding efficiency.

Table 1 shows the statistics of the number of tokens processed by different tokenizers[8]. The most efficient tokenizers for Chinese are those of Chat-GLM and BLOOMZ, indicated by the medians of token numbers. In contrast, the tokenizer of ChatGPT produces 2× tokens and that of Vicuna produces 2.5× tokens. The truncation length is proper to accommodate most samples.

## 4 LLM vs. LLM with IR System

We initially present the overall results, highlighting the importance of label candidates and similar cases, and conduct a comparative analysis of the models. Subsequently, we investigate the relationship between label candidates and self-consistency to unveil their actual effects on expertise reasoning. Additionally, we perform an ablation study by replacing similar cases with fixed cases as demonstrations to further understand their impact.

### 4.1 Overall Results

The macro comparison between the four settings is shown in Figure 2, where each point represents the performance of one specific run of one model.

**Significance of label candidates and similar cases.** In comparison to the zero-shot open question setting where LLMs work alone, the inclusion of label candidates, similar cases, or both demonstrates noteworthy enhancements. This highlights the effectiveness of our baseline solutions that leverage IR systems to expand the capabilities of LLMs in legal domains. These findings align with previous research that has also recognized the significance of the two components (Ma et al., 2023; Liu et al., 2021).

The effects of label candidates and similar cases differ slightly in terms of performance mean and

---
[8]GPT-4 and ChatGPT have the same results. Following OpenAI's guidance, we use the python package `tiktoken` for tokenization

variance. Label candidates contribute to a higher mean performance, while similar cases introduce greater variance. Examining the model performances in the third setting (+Sim Case) displayed in Figure 2, GPT-4 and ChatGPT exhibit more significant improvements from similar cases compared to their smaller counterparts. They also gain more benefit from similar cases than from label candidates. This observation can be attributed to the varying difficulty levels of knowledge utilization. While the knowledge within label candidates is readily accessible and straightforward, leveraging similar cases requires stronger language understanding and few-shot learning abilities.

Furthermore, the coexistence of label candidates and similar cases further enhances the performance of GPT-4 and ChatGPT, but it diminishes the performance of Vicuna, ChatGLM, and BLOOMZ. This suggests that smaller LLMs may encounter challenges in effectively managing knowledge in multiple forms simultaneously, leading to confusion.

**Model comparison.** The performances of the models under zero-shot and few-shot prompting is shown in Figure 3, where few-shot performances are averaged among 1-shot to 4-shot.

The zero-shot setting emphasizes the ability to understand instructions. When only instructions are available, BLOOMZ performs better than Chat-GPT, indicating a superior multilingual instruction following ability. This result is reasonable as BLOOMZ is the only smaller LLM that is fine-tuned on multilingual instructions. Once provided with explicit domain knowledge, ChatGPT outperforms all smaller LLMs. The case is the same for BLOOMZ and ChatGLM, where Chat-GLM overtakes BLOOMZ with knowledge of label candidates. BLOOMZ performs worst when prompted with two forms of knowledge, indicating that BLOOMZ is not very robust to prompts. Among the three smaller LLMs, ChatGLM is the most robust to various forms of knowledge.

The significant effects of label candidates and similar cases can be explained as they activate LLM's memory of relevant domain knowledge. This view can be supported by two pieces of evidence about the relationship between label candidates and self-consistency (Section 4.2) and the negligible effect of irrelevant cases as fixed demonstrations (Section 4.3).

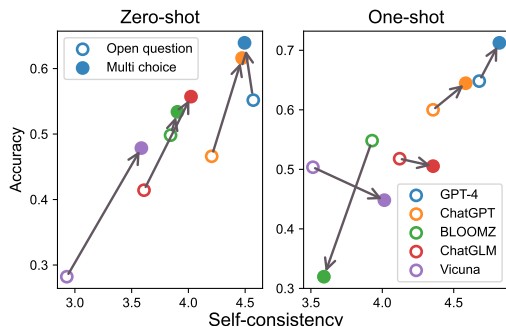

Figure 4: Changes of performance and self-consistency after adding label candidates. The change of each model is illustrated by an arrow pointing from the open question setting to the multi-choice setting.

## 4.2 Label Candidates Enhance Self-consistency and Confidence

To further understand the effect of label candidates, we propose a metric to measure the self-consistency of LLMs that is calculated as the number of the majority prediction[9]. Consistent outputs indicate a high level of confidence in LLMs, which is often associated with a better grasp of knowledge (Jiang et al., 2021, 2023).

The changes in performance and self-consistency after introducing label candidates are shown in Figure 4 as the arrows. We observe that the incorporation of label candidates leads to more consistent outputs (8 of 10 cases) and higher confidence in LLMs except zero-shot GPT-4 with a slight decrease and few-shot BLOOMZ. In the zero-shot setting, label candidates significantly boost LLM performances. We postulate that label candidates help by eliciting pre-stored domain knowledge with concise charge names. Besides, the self-consistency also correlates with model performances (7 of 10 cases). Such correlation is also observed in other tasks like question answering (Jiang et al., 2021). It is worth noting that label candidates decrease both self-consistency and performance of few-shot prompted BLOOMZ, which also aligns with the correlation.

## 4.3 Domain Knowledge Is More Critical Than Task Illustration

There is a possible argument that similar demonstrations can help LLMs understand instructions and tasks. To disentangle their effects on task illustration and provision of domain knowledge, we

---

[9]For example, if the five sampled outputs are mapped to labels of (a,a,a,b,c), the consistency score is 3.

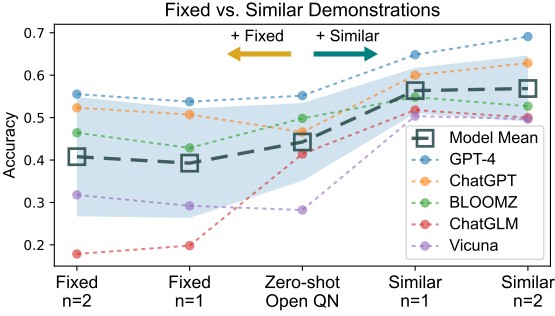

Figure 5: The effects of fixed (irrelevant) and similar cases as demonstrations. Divided by the baseline setting of zero-shot open questions, the left part refers to fixed demonstrations with increasing numbers of demonstrations, while the right part refers to similar demonstrations. The shadow area represents the range of standard deviation.

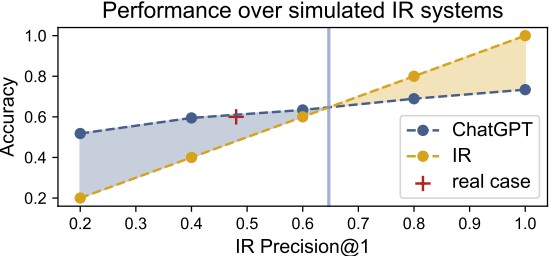

Figure 6: The performance of ChatGPT coordinated with a series of simulated IR systems with varying capabilities as measured by Precision@1. The vertical blue line represents the threshold of IR capability at which IR systems overtake ChatGPT. The performance of ChatGPT in the real setting (1-shot open questions) is indicated by the red plus sign.

experiment with irrelevant demonstrations fixed for all test samples. We manually select two common cases with frequent charges in the original dataset as the fixed demonstrations. The 1-shot performance was averaged on the two demonstrations.

We compare the effects of fixed and similar demonstrations with the baseline setting of zero-shot open questions in Figure 5. The change of performance from center to left demonstrates that fixed demonstrations hardly benefit LLMs and sometimes harm the performance (*e.g.,* ChatGLM). This indicates that LLMs can basically understand instructions and do not need general demonstrations for task clarification, implying that the main challenge of expertise reasoning is to recall domain knowledge instead of understanding a specific task.

We inspect the notable performance drop of ChatGLM resulting from fixed demonstrations. We find that ChatGLM tends to analyze the cases of both demonstrations and test samples and then answer with both of their charges. Its wordy style seems to result from the fine-tuning dialog corpus where an assistant LLM is supposed to provide rich information. In contrast, similar cases seem to encourage more concise outputs following the format of demonstrations.

## 5 Paradox of Information Retrieval System

The significance of similar demonstrations illustrated in Section 4.3 has motivated research focusing on prompting-oriented IR systems (Rubin et al., 2021; Sun et al., 2023) to retrieve high qual-

ity demonstrations. However, we raise an intuitive question: *Do LLMs gain substantial improvement from IR systems compared to the kNN baseline that harnesses IR systems for classification tasks*? The question is inspired by our observation that the BM25 retriever achieves 48.03% of Precision@1 [10] and 57.68% prediction accuracy by majority vote of top $k = 17$ retrieved similar cases.

This observation suggests a **paradoxical scenario** wherein an IR system outperforms the combination of LLM and IR, with the LLM taking on the leading role and the IR serving as a supporting role. In such a scenario, the LLM becomes redundant due to its failure to fully utilize the informative retrieved documents.

To investigate the paradox, instead of experimenting with different IR systems, we manipulate the BM25 retriever to simulate a series of IR systems with different capabilities measured by Precision@1 as described by Section 2.3. We take a case study of ChatGPT, whose 1-shot performance under different IR systems (denoted as Precision@1) is shown in Figure 6.

**Results** Although the performance of ChatGPT enhanced by IR systems improves with IR capability, it will eventually underperform the IR system once the IR capability surpasses a certain threshold. In the ideal situation where true similar cases are always retrieved, ChatGPT is unable to attain 100% accuracy and lags significantly behind the optimal IR system. According to Appendix A.4, all smaller LLMs are not comparable to the BM25 retriever.

**Discussion** The findings demonstrate that LLMs face challenges in effectively leveraging informa-

---

[10]It is identical to the precision of $k$NN with $k = 1$

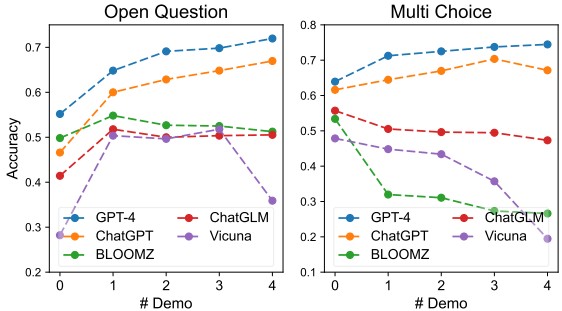

Figure 7: Performance vs. the number of similar demonstrations of the five LLMs.

tive retrieved documents. This underscores the need for significant research efforts to enhance the synergy between auto-regressive language models and retrieval by conditioning model outputs more on retrieved documents. Previous work has explored the augmentation of LLMs with retrieval at both the pre-training and fine-tuning stages (Borgeaud et al., 2022; Wang et al., 2023). Moreover, the marginal and inadequate improvement with retrieval indicates the limited legal reasoning ability of existing general LLMs. There is a need for future efforts to enhance domain-specific reasoning abilities of pre-trained foundation models.

## 6 Ablation Study

### 6.1 More Demonstrations Are Not Always Better

The impact of the number of similar demonstrations ($n$) is depicted in Figure 7. It is evident that GPT-4 and ChatGPT demonstrate proficiency in handling larger numbers of demonstrations, leading to enhanced performance, whereas Vicuna, Chat-GLM and BLOOZ experience varying degrees of performance degradation with increasing numbers. Notably, ChatGLM displays the least sensitivity to $n$. Furthermore, even ChatGPT's performance declines when $n$ is increased from three to four. The performance improvement resulting from larger values of $n$ can be attributed to the increased recall of true similar cases. Conversely, the decline in performance can be attributed to the noise introduced by more false similar cases.

**Performance variations.** The change of performance after including an additional demonstration are visualized using heat maps in Figure 8. For each model, the three heat maps stand for the variations from k-shot to (k+1)-shot, which are denoted below. For each heat map, the two rows indicate

the inclusion of a new demonstration with true (T) or false (F) similar cases, while the columns indicate the combinations of existing demonstrations. Take the second heat map as an example. The cell in the column of (F, T) and the row of (T) displays the performance variation between 2-shot of (F, T) demonstrations and 3-shot of (F, T, T) demonstrations. Purple represents performance improvement, while green represents performance decline.

For ChatGPT and BLOOMZ, the second rows of their three heat maps are mainly in purple, indicating significant enhancements resulting from the inclusion of true similar cases. However, the first lines of BLOOMZ display a deeper green color than those of ChatGPT, suggesting that BLOOMZ experiences greater degree of performance declines caused by the inclusion of false similar cases. These findings indicate different sensitivity to false similar demonstrations. Powerful language models like GPT-4 and ChatGPT exhibit robustness to noise in false similar cases, allowing them to remain focused on relevant information in true similar cases. In contrast, weaker LLMs are susceptible to the influence of such noise. Overall, ChatGPT performs better when provided with more similar demonstrations, whereas BLOOMZ demonstrates the opposite, as shown in Figure 7.

The conclusion is that increased numbers of demonstrations have both positive and negative implications for expertise reasoning. However, LLMs could potentially gain from additional demonstrations in tasks that requires clear task illustration.

### 6.2 The Impact of Absent Ground Truth Labels

We manually incorporate ground-truth labels into label candidates in cases where they are absent, which may occur due to the limited recall capability of the IR system described in Section 2.2. The test samples are categorized into two groups, namely "Easy" and "Hard", based on the retrieval of their ground truth labels by the IR system. The original performance of the two groups and the performance of the "Hard" group with modified prompts to include ground truth labels, namely "Hard+GT", are displayed in Figure 9.

The performance gaps between the "Easy" and "Hard+GT" groups suggest that challenging samples for IR systems are also difficult for LLMs. However, this gap is insignificant for the powerful GPT-4 who perceives them as equal challeng-

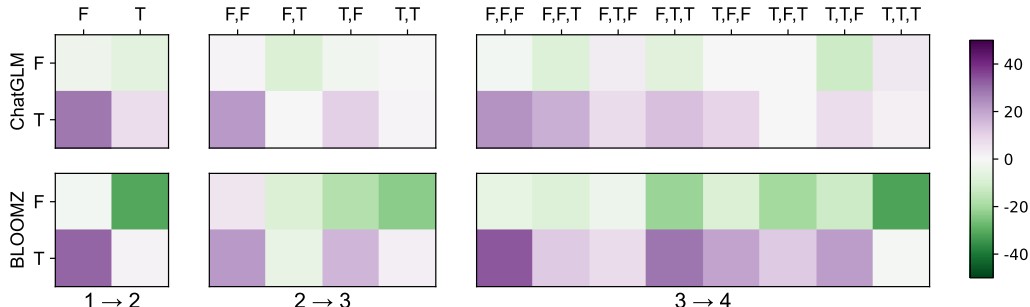

Figure 8: Heat maps of performance variations resulting from the inclusion of an addition demonstration. "T" corresponds to demonstrations with true similar cases, while "F" represents those with false similar cases. Each row represents the included new demonstration, while each column indicate the status of existing demonstrations.

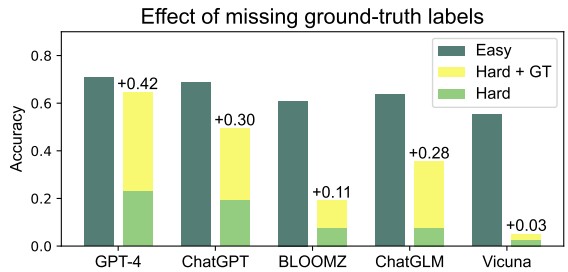

Figure 9: The performance of "Easy" and "Hard" samples under the setting of zero-shot multi-choice questions. "Hard+GT" refers to improvement of including the absent ground truth labels in label candidates.

ing. The improvement of "Hard+GT" compared to "Hard" is notable in GPT-4, ChatGPT and ChatGLM but inconspicuous in Vicuna with inferior competency in the law. Considering the relatively small size of the "Hard" group (79/560), the absence of ground truth labels does not have a significant impact, especially for weaker LLMs.

### 6.3 Incorporation of Law Articles

We examine the effect of incorporating legal articles that explicitly define the charges into prompts. For each charge retrieved by the IR system[11], ChatGPT is required to determine whether the defendant is guilty for the particular charge by answering with a yes or no. We find that 94.46% of the ground truth charges are accurately detected, while only 27.31% of the detected charges are correct. The high recall and low precision indicate a substantial difference between ChatGPT and legal experts in the ability to distinguish charges and make precise judgments.

---

[11]we also include the ground truth charge

## 7 Discussion

We compare the LLMs with supervised baselines. We fine-tune BERT (Devlin et al., 2018) on the same training set and achieve a comparable accuracy of 68% to ChatGPT but lower than GPT-4. Since LLMs are not fine-tuned on the specific LJP task, this result highlights the remarkable superiority of LLMs in acquiring significant knowledge and leveraging transfer learning Raffel et al. (2020).

However, we observe that BERT's performance improves to 89% when trained with the original training set (~10K). We find that certain knowledge is present in shadow features, which can be easily learned with supervision. These superficial features can result in biased supervised models. Fortunately, unsupervised pre-training objectives, make LLMs more robust and less vulnerable to this issue. This depicts a promising future for NLP applications in various domains.

## 8 Conclusion

To address the deficiency in evaluating the competency of LLMs in the field of law, we focused on the task of legal judgment prediction and devised four settings to facilitate a thorough evaluation that encompassed both open and multiple-choice questions and incorporated similar cases to aid in the decision-making process.

The evaluation results revealed different behaviors among the prominent LLMs, namely GPT-4 and ChatGPT, compared to their smaller counterparts. Both GPT-4 and ChatGPT exhibited remarkable proficiency in effectively leveraging domain knowledge in various formats. Among the smaller LLMs, ChatGLM displayed greater robustness, while BLOOMZ showcased superior zero-shot ability.

We presented an intriguing paradox wherein LLMs could become abundant in the presence of a powerful IR system. When improving IR systems to benefit LLMs, it is crucial for researchers to acknowledge this paradoxical scenario and prevent great disparity between LLMs and IR systems.

## Limitations

One limitation of this paper is the use of the close-source GPT-4 and ChatGPT whose availability depends on the commercial company OpenAI. According to OpenAI, the ChatGPT and GPT-4 versions used in this paper, namely gpt-3.5-turbo-0301 and gpt-4-0314, will be deprecated and not available after September 13th, 2023.

Another limitation pertains to the selection of LLMs. Due to the rapid emergence of new LLMs, we are not able to include all of them with the constraint of limited time. Instead of more models, we focus more on designing comprehensive evaluation settings and conducting insightful analyses to shed light on other domains.

## Ethics Statement

The task of legal judgment prediction is used to evaluate LLM's competency in the law. The primary objective of this task is to assist judges and lawyers in comprehending lengthy legal documents by offering them a supplementary tool. It is important to note that this task does not seek to replace the roles of judges and lawyers, nor does it aim to determine the guilt or charges of defendants through machine learning algorithms. Additionally, there is research focused on interpreting LJP models, aiming to enhance the transparency of black-box models for improved utilization by legal practitioners. The paper utilizes a public and anonymized dataset to exclude the potential issue of personal information leakage.

## Acknowledgements

We thank all reviewers for their constructive comments. This research is supported by NExT Research Center, the National Natural Science Foundation of China (9227010114) and the University Synergy Innovation Program of Anhui Province (GXXT-2022-040).

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

# A Appendix

## A.1 Prompt Templates

The prompt template is shown in Figure 10. The translation of the original Chinese prompt is displayed using orange text. The setting of zero-shot open questions use a longer instruction that appends "Output the charge name directly" to the instruction in Figure A.1.

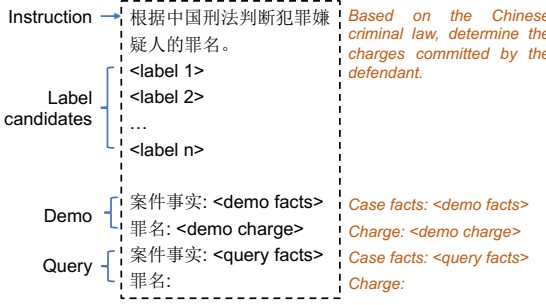

Figure 10: The prompt template in Chinese and English.

## A.2 Robust to Fixed Demonstrations

| Model | 1shot | 2shot |
|---|---|---|
| GPT-4 | 49.59 / 48.84 | 50.69 |
| ChatGPT | 47.01 / 46.57 | 47.55 |
| Vicuna-13B | 22.74 / 29.38 | 28.37 |
| ChatGLM-6B | 22.39 / 25.14 | 21.36 |
| BLOOMZ-7B | 36.65 / 43.94 | 42.24 |

Table 2: The classification accuracy scores with prompts consisting of fixed cases.

We examine the effects of the two fixed cases mentioned in Section 4.3 in Table 2. We find that GPT-4 and ChatGPT are robust to the selection of the fixed demonstration in 1-shot setting, while Vicuna, ChatGLM and BLOOMZ are less robust.

## A.3 Comparison with Supervised Baselines

To understand the performance of supervised fine-tuning (SFT) baselines on LJP, we experiment on three models: BERT[12], XLM-RoBERTa[13] and DeBERTa[14]. These models are fine-tuned on two datasets of different sizes: the original CAIL dataset (~100k samples) and the sampled training set (1120 samples) that is used as retrieval corpus described in Section 3.2, denoted as CAIL_few.

---

[12]bert-base-chinese
[13]xlm-roberta-base
[14]microsoft/mdeberta-v3-base

---

The SFT models are evaluated on the same evaluation dataset described in Section 3.2. The smaller training set aims to compare the few-shot performance of SFT baselines and LLMs in low data scenario.

The results of SFT models are shown in Figure 3. Considering the highest accuracy of GPT-4 being 74.46% (multi-choice, 4shot), GPT-4 can outperform supervised baselines in low data scenario. If there is abundant training data, supervised baselines are still better than GPT-4 by 15%.

| Model | CAIL | CAIL_few |
|---|---|---|
| BERT | 89.64 | 68.04 |
| XLM-RoBERTa | 88.75 | 66.43 |
| DeBERTa | 88.57 | 30.89 |

Table 3: Prediction accuracy of SFT models fine-tuned on two training datasets of different sizes.

## A.4 Detailed Results

The specific values of performances displayed in Figure 2 are presented in Table 4. Besides, we also provide the performance of the F1 score in Table 5.

| Model | Open Questions | | | | | Multiple-choice Questions | | | | |
|---|---|---|---|---|---|---|---|---|---|---|
| | 0shot | 1shot | 2shot | 3shot | 4shot | 0shot | 1shot | 2shot | 3shot | 4shot |
| GPT-4 | 55.18 | 64.82 | 69.11 | 69.82 | 71.96 | 63.93 | 71.25 | 72.50 | 73.75 | 74.46 |
| ChatGPT | 46.61 | 60.00 | 62.86 | 64.82 | 66.96 | 61.61 | 64.46 | 66.96 | 70.36 | 67.14 |
| Vicuna-13B | 28.21 | 50.36 | 49.64 | 51.79 | 35.89 | 47.86 | 44.82 | 43.39 | 35.71 | 19.46 |
| ChatGLM-6B | 41.43 | 51.79 | 50.00 | 50.36 | 50.54 | 55.71 | 50.54 | 49.64 | 49.46 | 47.32 |
| BLOOMZ-7B | 49.82 | 54.82 | 52.68 | 52.50 | 51.25 | 53.39 | 31.96 | 31.07 | 27.32 | 26.61 |

Table 4: The classification accuracy scores of all models under the four settings.

| Model | Open Questions | | | | | Multiple-choice Questions | | | | |
|---|---|---|---|---|---|---|---|---|---|---|
| | 0shot | 1shot | 2shot | 3shot | 4shot | 0shot | 1shot | 2shot | 3shot | 4shot |
| GPT-4 | 50.52 | 62.72 | 67.54 | 68.61 | 71.02 | 62.31 | 70.42 | 71.81 | 73.24 | 74.00 |
| ChatGPT | 43.14 | 58.42 | 61.86 | 64.40 | 66.16 | 60.67 | 63.51 | 66.85 | 69.59 | 66.62 |
| Vicuna-13B | 25.50 | 48.85 | 47.64 | 49.49 | 39.82 | 44.70 | 41.73 | 41.48 | 35.03 | 21.61 |
| ChatGLM-6B | 41.89 | 50.30 | 47.76 | 48.59 | 48.67 | 53.74 | 49.26 | 47.56 | 47.61 | 45.32 |
| BLOOMZ-7B | 46.90 | 53.28 | 51.06 | 50.90 | 49.26 | 50.68 | 29.25 | 27.92 | 25.27 | 23.37 |

Table 5: The classification F1 scores of all models under the four settings.