# OpenReview forum: "A Comprehensive Evaluation of Large Language Models on Legal Judgment Prediction"
_EMNLP/2023/Conference — EMNLP 2023 Findings_

### Official Review · Reviewer_9huf · 2023-08-03

**Typos Grammar Style And Presentation Improvements:** line 430
**Soundness:** 4

**Excitement:**

3: Ambivalent: It has merits (e.g., it reports state-of-the-art results, the idea is nice), but there are key weaknesses (e.g., it describes incremental work), and it can significantly benefit from another round of revision. However, I won't object to accepting it if my co-reviewers champion it.

**Paper Topic And Main Contributions:**

The paper is centered on assessing the performance and utility of large language models (LLMs) in the legal domain, specifically focusing on legal judgment prediction. This paper provides a comprehensive evaluation of LLMs in legal judgment prediction tasks. This evaluation involves designing and comparing various models or approaches, including GPT-4, in terms of their performance on legal judgment prediction. This can help to clarify the strengths and limitations of using LLMs in the legal domain.

**Reasons To Accept:**

1. The paper did a pretty detailed and comprehensive study on LLM's ability of legal domain using various prompting methods.
2. The paper discovers the interesting observation that well-performing IR can perform better than LLM + IR.
3. very nice graphs and clear demonstrations

**Reasons To Reject:**

1. The paper investigates a lot of the effects of prompting methods. But I hope the authors can compare the result with general domain prompting results, as (1) many conclusions seem to overlap with general domain prompting results and (2) authors should highlight the specific/new difficulty met in the legal domain, thus rendering the study less interesting.
2. The section 3.2, the paper mentions the difference in tokenizers, which is very interesting. I would like to see how the tokenizer difference may explain/correspond to some experiment phenomena, but unfortunately, it is not discussed.


**Reproducibility:**

4: Could mostly reproduce the results, but there may be some variation because of sample variance or minor variations in their interpretation of the protocol or method.

**Reviewer Confidence:**

4: Quite sure. I tried to check the important points carefully. It's unlikely, though conceivable, that I missed something that should affect my ratings.

---

> ### Author Rebuttal · Authors · 2023-08-29
>
> Thanks for your recognition of our work. We are pleased to receive your critical comments. Our evaluation benchmark considers the scenario of LLM + retrieval system, which is a recent hot topic of retrieval augmented LLMs (ACL 2023 Tutorial: Retrieval-based Language Models and Applications). As a legal benchmark, we showcase the effects of two types of legal knowledge: charge names and similar cases (precedents), on legal judgment prediction. Besides, the observed “paradox” indicates the importance of retrieval quality and suggests two future directions of LLM + IR: 1) how to pretrain LLMs to ignore retrieval noise (Wang et al. 2023) and 2) how to enhance retrieval systems to deliver user-friendly information to LLMs.
>
> Due to the limitation of space, some results are not included in the paper. We would like to explain with more experiment results, examples, and statistics and hope they can answer your constructive questions to some extent.
>
> ## Res to Reject Reason 1
> We are interested into how in-context learning works and have conducted some ablation study on ChatGPT. Besides, we also manually check the LLM outputs to identify the characteristics of challenging cases.
> ### Corrupted Demonstrations in In-context Learning
> To explore to what extent LLMs can learn from in-context demos, we corrupt and build artificial demos by two strategies. In the first strategy, the demo input text (i.e., case description) is chosen from a random case, while the output (i.e., charge label) is set to be identical to the query case, denoted as **random input+ground label**. In the second strategy, we choose a true similar case as input text while set the output label to be random, demoted as **misleading**. We experiment on ChatGPT and compare the two artificial demo strategies to the **oracle**, where the demos are true similar cases without any corruption (i.e., cases retrieved by a 100% accurate retrieval system). The results are:
> | Demo Strategy | Acc |
> | - | - |
> | Oracle | 73.39 |
> | Random input + Ground label | 65.89 |
> | Misleading | 51.61 |
>
> The results show that even the input-output correlation is broken, the model’s performance only drops slightly and it can still provide correct answers in many cases. It suggests that the model do not totally rely on the context to make predictions and mostly rely on its **prior knowledge** acquired during pretraining. Previous work of demystifying in-context learning also has similar observations (Min et al. 2022, Wei et al. 2023(a), Wei et al. 2023(b)).
>
> Besides, the “Ground label” setting yields a much higher accuracy than the “Misleading” setting. This suggests that the ground truth labels existing in prompts can provide strong hints for LLMs to reach the correct predictions. We conjecture that ground truth labels, or *label candidates* described in the paper, can activate LLM’s memory of relevant domain knowledge.
>
> ### Case study
> Besides the confusing charges, LJP is challenging to both LLMs and supervised models as there could be **multiple criminal actions/facts in one case**. For example, one defendant may rob the handbag of the victim, and use the victim’s credit card to withdraw money. The lexical features can be correlated to the labels of both robbery and credit card fraud, while legal experts can tell that the proper charge should be robbery.
>
> Another case is that when multiple laws are violated, the ultimate judgment should be based on the charge with the most stringent conditions and harsher penalties. For example, “Affray” could escalate to the graver offense of “Intentional Injury” if the injuries of the victims are deemed severe.
>
> We have observed that small LMs, e.g., BERT, Roberta and DeBERTa, finetuned on the task specific data would **fail** these examples due to weaker reasoning ability and relying on superficial features. LLMs with standard prompting methods can also make mistakes when handling these cases, but they are promising to be competent if prompted properly. There have been increasing attempts to harness LLMs to solve complex reasoning tasks, e.g., task decomposition, improving consistency, embodied agents, etc.
>
>
> ## Res to Reject Reason 2
> Your question also interests us a lot. The main effect of tokenizers seems to be the different tokenization efficiency for different language, which results from the distribution of pretraining corpus. The resulting different context length may pose the challenge for LMs to capture long term dependency. We are not experts of this domain, so we try to give some empirical statistics.
>
> For each model, we aim to display the relationship between the prompt length and prediction accuracy. We sort all samples based on the numbers of prompt tokens and equally split them into 5 groups. That is, the token number upper limits of these groups are the 20, 40, 60, 80, 100 percentiles. The performances (accuracy) of the samples of the groups are averaged among 1-shot to 4-shot settings, and we calculate the performance drop of the longest group compared to the shortest group. Below are the results:
> | model | 1(short) | 2 | 3 | 4 | 5(long) | drop |
> | - | - | - | - | - | - | - |
> | gpt4 | 75.67 | 78.84 | 71.72 | 61.52 | 56.83 | -18.84 |
> | chatgpt | 71.43 | 72.16 | 68.61 | 55.27 | 50.80 | -20.63 |
> | bloomz | 63.18 | 61.88 | 54.99 | 46.32 | 37.57 | -25.61 |
> | chatglm | 61.48 | 60.81 | 53.15 | 41.74 | 36.10 | -25.38 |
> | vicuna | 61.83 | 61.24 | 54.34 | 32.69 | 24.33 |  -37.5 |
>
> The results indicate that all models experience significant decrease of performance on longer prompts. There could be another confounder that longer samples are inherently difficult. However, the varying degrees of performance drop suggest LLMs different abilities to handle long inputs (around 2000 tokens in our experiments). If we take GPT-4 as a baseline and attribute its performance drop to the inherent difficulty of long samples, the additional ~8% drop of Bloomz and ChatGLM can be attributed to their inferior ability to model long sequences.
>
> In our experiment settings, we truncate the query case to 1000 tokens that can preserve most of the information (see Table 1). We also check this by feeding Vicuna’s prompts (truncated by the least efficient tokenizer) to ChatGLM and Bloomz, and we do not observe significant performance changes compared to prompts truncated by their own tokenizers. This indicates that in our experiments, where we prioritize the truncation of demonstrations and the maximum shot is set to 4, useful information is well maintained and the tokenizer efficiency is not the bottleneck of model performance.
>
> **Many thanks for your insightful feedback and your time on our rebuttal.**
>
> ### References
> - Shall We Pretrain Autoregressive Language Models with Retrieval? A Comprehensive Study. Wang et al. 2023.
> - Rethinking the Role of Demonstrations: What Makes In-Context Learning Work? Min et al. 2022 EMNLP
> - Larger language models do in-context learning differently. Wei et al., 2023 (a)
> - Symbol tuning improves in-context learning in language models. Wei et al., 2023 (b)

---

### Official Review · Reviewer_JuZH · 2023-08-03

**Soundness:** 3

**Excitement:**

3: Ambivalent: It has merits (e.g., it reports state-of-the-art results, the idea is nice), but there are key weaknesses (e.g., it describes incremental work), and it can significantly benefit from another round of revision. However, I won't object to accepting it if my co-reviewers champion it.

**Paper Topic And Main Contributions:**

This paper evaluates several LLM on a specific task in the legal domain.  The specific task is to map a fact description to a specific legal charge.  (The example in Fig 1 is helpful:  "A purse was stolen" -> "theft").  Two types of experiments are run. One uses just LLM.  The second uses LLM's and an Information Retrieval system to provide the LLM with a similar case.  The results rank the tested LLMs based on accuracy with GPT-4 ranking best.



**Questions For The Authors:**

A:  Perhaps I am not familiar enough with this dataset.  I would appreciate more detail on the labels that are used.  In particular, is there a natural metric space where some labels are close.  Eg., Theft vs Grand Theft.  Murder vs homicide.

B: Are the model errors (or corrects) correlated across models?  Is GPT-4 better in the sense that it gets the same things correct as other models and then additional corrects as well?

C: I found Figure 8 hard to interpret. What should be my focus when I look at that picture?  Is it the relative color (performance) in the same position across the models?

**Reasons To Accept:**

- The paper is very clear/clean, setting to test LLMs.  The task uses clear and well-labeled data.
- The experiments are straightforward, so it is easy (and fair) to compare the performance of different model

**Reasons To Reject:**

- While this is a clear/clean setting for evaluation, I could not tell from the paper if this a practical and important use case for law.  Is correctly labeling situations an important goal?  Or, is the argument that LLMs have to do well on this simpler task before they can be useful in more complex tasks (like framing the charges or defense)?

- The "paradox" that an information retrieval system may or may not help performance is interesting, but the implication for research is unclear.  The conclusion: "The conclusion is that increased numbers of demonstrations have both positive and negative implications for expertise reasoning."  This does not seem to illustrate why or what to do next.

**Reproducibility:**

4: Could mostly reproduce the results, but there may be some variation because of sample variance or minor variations in their interpretation of the protocol or method.

**Reviewer Confidence:**

3: Pretty sure, but there's a chance I missed something. Although I have a good feel for this area in general, I did not carefully check the paper's details, e.g., the math, experimental design, or novelty.

**Typos Grammar Style And Presentation Improvements:**

As a general suggestion, shorter sentences might make things easier for the reader.  If an example would help:

from the paper:
"Given that label candidates only hint LLMs with limited information of charge names, it is more possible that the significant improvement is attributed to the recall of pre-stored domain knowledge, which is elicited by label candidates."

As two sentences:
The label candidates help LLM performance.  We postulate that the label candidate helps by eliciting pre-stored domain knowledge.

---

> ### Author Rebuttal · Authors · 2023-08-28
>
> We would like to express our sincere gratitude for your insightful feedback and constructive comments and thanks for your efforts on reviewing our paper. We are encouraged by your appreciation on our “clear, fair and straightforward” evaluation benchmark. We are going to public the toolkit to evaluate LLMs by one-line command and maintain a leaderboard of legal LLMs.
>
> Besides the merits, we are also pleased to hear your questions and concerns that are helpful for improving our paper. We hope the response bellow can provide a clearer explanation of our work.
>
> ### Res to Quest. A:
> The labels (charges) covered by the dataset are diverse while there are also close and confusing ones. We provide some statistics and analyze the ChatGPT results to showcase confusion labels.
>
> The labels are the most common **112 charges** of the criminal law of China, which are categorized into **10 chapters**. Below are the most common chapters, number of included charges and charge examples:
> - *Crime of disrupting social order* (35), e.g., Affray, Environmental Pollution, and Illegal Possession of Drugs
> - *Crime of Disrupting the Socialist Market Economy Order* (31), e.g., Producing or Selling Food Products that Do Not Meet Safety Standards and Contract Fraud.
> - *Crime of Violating Personal and Democratic Rights* (17), e.g., Intentional Injury
> - *Crime of Endangering Public Safety* (15), e.g., Dangerous Driving and Arson.
> - *Crimes Against Property* (9), e.g., Fraud and Forcible Seizure
>
> The confusing charges are determined by the confusion matrix of ChatGPT. We build the graph of charges (nodes) and link two charges if the counts of misclassification between the two charges exceed the threshold 7, based on the results described in Section 6.3. The charges in a *maximal connected subgraph* are recognized as confusing charges. Below are some confusing charge groups:
> - Producing or Selling Food Products that Do Not Meet Safety Standards; Producing or Selling Toxic or Harmful Food; Producing or Selling Counterfeit Drugs
> - Affray; Illegal Detention; Intentional Injury
> - Negligently Causing Serious Accident; Negligence Causing Death; Dangerous Driving; Traffic Accident
>
> ### Res to Quest. B
> To investigate the correlations of two models and the randomness of results, we propose two metrics
> -  *Agree*: to measure the ratio of **agreeing** cases where two models output same predictions
> - *Gain*: to measure the ratio of cases that the first model is correct if two model **disagrees**
>
>  We divide the samples into four groups based on the correctness of two models, denoted as {T, F}. For example, when comparing GPT-4 and BLOOMZ, the *TF* group means that GPT-4 gets a correct result while BLOOMZ fails. Then the two metrics are calculated as
> $$Agree=\frac{TT+FF}{TT+TF+FT+TT}$$
> $$Gain=\frac{TF}{TF+FT}$$
>
> The open question results of zero shot and few-shot are (compare GPT-4 to other models):
> | zero-shot | ChatGPT| ChatGLM | BLOOMZ| Vicuna|
> |-|-|-|-|-|
> | Agree | 78.93 | 70.18 | 75.00 | 63.04 |
> | Gain | 70.34 | 73.05 | 60.71 | 86.47 |
>
> | 2-shot | ChatGPT| ChatGLM | BLOOMZ| Vicuna|
> |-|-|-|-|-|
> | Agree | 83.75 | 68.75 | 70.71 | 71.61 |
> | Gain | 69.23 | 80.57 | 78.05 | 84.28 |
>
> As GPT-4 benefits more from similar cases (Section 4.1, Line 301-303), the *Gain* under 2-shot setting over ChatGLM and BLOOMZ increases to around 80%, which means that when two model disagrees, GPT-4 can get correct predictions four fifths of the time, and the smaller counterparts is only correct one fifths of the time. This significant evidence shows that GPT-4 can do better on samples that other models fail on.
>
> Then we take a close look at the numbers of samples grouped by GPT-4 and ChatGLM results of 2-shot setting, which is {'TT': 246, 'TF': 141, 'FT': 34, 'FF': 139}. It shows that compared to ChatGLM, GPT-4 get additional 141(25.2%) samples correct and 34(6.1%) samples wrong, resulting in a 19.1% performance improvement (consistent to results in Table 2 in Appendix).
>
> As there may be some uncertainty in LM generation and result parsing, the above results show clear improvements of GPT-4 over smaller counterparts.
>
> ### Res to Quest. C
> We have tried several layouts of Figure 8 and we recognize that it may be challenging to comprehend. The focus should be the comparison of the **two horizon blocks (Green vs. Purple)**, denoted as “F” and “T” at the left, of **each model**. The “F” line represents the performance changes after including a false similar case into a prompt, and its “*Green*” color means that the performance decreases due to the new false similar case. In contract, the “T” line featured by the “*Purple*” color indicates a performance enhancement due to a new true similar case in prompts. The implication of Figure 8 will be explained later to address the Reject Reason 2.
>
> ### Res to Reject Reason 1
> Legal judgment prediction is a common task in Legal NLP, which is usually formulated as a classification task (Zhong et al. EMNLP 2018, Chalkidis et al. ACL 2019). LJP can serve as a fundamental task for other applications such as legal consultation, and the research of interpretable LJP can benefit both legal practitioners and laymen to comprehend lengthy legal documents (Wu et al. EMNLP 2022, Zhong et al. AAAI 2020).
>
> We select this task for evaluating LLMs as LJP requires multiple level of legal abilities, including the knowledge of laws and legal terms and legal expertise reasoning. Besides, the ground truth labels of LJP make it easy to evaluate the outputs of LLMs. As shown in the response to Question A, LJP remains challenging in discriminating between confusing charges, an issue that that legal experts strive to address.
>
> There are also other possible tasks to evaluate legal LLM, such as bar exam and legal document writing (e.g., case report, court debate and defense) as you suggested. The legal writing tasks are more challenging for requiring sophisticated logic and coherence. However, the main issue lies in the difficulty in content evaluation. The automatic evaluation metrics, e.g., BLEU and ROGUE, mainly highlight the text similarity and thus are not an appropriate measurement of knowledge and logic intense content. The alternative is human evaluation that was adopted by Open AI to evaluate the writing part in the bar exam of GPT-4, but it is costly, time-consuming and subject, hindering the reproduction of evaluation results.
>
> We acknowledge your viewpoint “*LLMs have to do well on this simpler task before they can be useful in more complex tasks*” as both objective and insightful, which motivates us to think deeply and systematically investigate this question in our future work.
>
> ### Res to Reject Reason 2
> Thanks for pointing out the unclear writing of the conclusion. The paradox and the ablation study of the number of similar cases indicate the importance of retrieval systems for LLMs. Recently, retrieval-based (augmented) LMs are drawing increasing attention (ACL 2023 Tutorial https://2023.aclweb.org/program/tutorials/).
>
> Our results point out two research directions: 1) how to pretrain LLMs to work with retrieval noise (Wang et al. 2023) and 2) how to improve the capability of retrieval systems to provide useful information for LLMs, e.g., searching from structured data like knowledge graph.
>
> ### Conclusion
> Finally, **thanks again for your review and the writing suggestions** that are very helpful to improve our paper. Limited by our abilities and time, we only choose one task to build the benchmark, but we try to make it a comprehensive one by including multiple fine-grained settings to showcase the two abilities of LLMs: using domain knowledge (multi-choice) and learning from previous cases (few-shot). As one of the pioneering legal and domain-specific LLM benchmarks, our framework and analyses can facilitate the proposal of more legal LLM benchmarks, and the settings tailored for LMs can also be transferred to other domains, such as medicine and finance.
>
> ### Reference
> - Legal Judgment Prediction via Topological Learning. Zhong et al. EMNLP 2018
> - Neural legal judgment prediction in English. Chalkidis et al. ACL 2019
> - Towards Interactivity and Interpretability: A Rationale-based Legal Judgment Prediction Framework. Wu et al. EMNLP 2022
> - Iteratively Questioning and Answering for Interpretable Legal Judgment Prediction. Zhong et al. AAAI 2020

---

### Official Review · Reviewer_gyVE · 2023-08-06

**Soundness:** 3

**Excitement:**

3: Ambivalent: It has merits (e.g., it reports state-of-the-art results, the idea is nice), but there are key weaknesses (e.g., it describes incremental work), and it can significantly benefit from another round of revision. However, I won't object to accepting it if my co-reviewers champion it.

**Missing References:**

@inproceedings{DBLP:conf/acl/ChalkidisAA19,
  author       = {Ilias Chalkidis and
                  Ion Androutsopoulos and
                  Nikolaos Aletras},
  editor       = {Anna Korhonen and
                  David R. Traum and
                  Llu{\'{\i}}s M{\`{a}}rquez},
  title        = {Neural Legal Judgment Prediction in English},
  booktitle    = {Proceedings of the 57th Conference of the Association for Computational
                  Linguistics, {ACL} 2019, Florence, Italy, July 28- August 2, 2019,
                  Volume 1: Long Papers},
  pages        = {4317--4323},
  publisher    = {Association for Computational Linguistics},
  year         = {2019},
  url          = {https://doi.org/10.18653/v1/p19-1424},
  doi          = {10.18653/v1/p19-1424},
  timestamp    = {Fri, 06 Aug 2021 00:41:04 +0200},
  biburl       = {https://dblp.org/rec/conf/acl/ChalkidisAA19.bib},
  bibsource    = {dblp computer science bibliography, https://dblp.org}
}

@inproceedings{DBLP:conf/aaai/ZhongWTZ0S20,
  author       = {Haoxi Zhong and
                  Yuzhong Wang and
                  Cunchao Tu and
                  Tianyang Zhang and
                  Zhiyuan Liu and
                  Maosong Sun},
  title        = {Iteratively Questioning and Answering for Interpretable Legal Judgment
                  Prediction},
  booktitle    = {The Thirty-Fourth {AAAI} Conference on Artificial Intelligence, {AAAI}
                  2020, The Thirty-Second Innovative Applications of Artificial Intelligence
                  Conference, {IAAI} 2020, The Tenth {AAAI} Symposium on Educational
                  Advances in Artificial Intelligence, {EAAI} 2020, New York, NY, USA,
                  February 7-12, 2020},
  pages        = {1250--1257},
  publisher    = {{AAAI} Press},
  year         = {2020},
  url          = {https://ojs.aaai.org/index.php/AAAI/article/view/5479},
  timestamp    = {Mon, 07 Mar 2022 16:57:39 +0100},
  biburl       = {https://dblp.org/rec/conf/aaai/ZhongWTZ0S20.bib},
  bibsource    = {dblp computer science bibliography, https://dblp.org}
}

**Paper Topic And Main Contributions:**

The paper "A Comprehensive Evaluation of Large Language Models on Legal Judgment Prediction" addresses the Legal Judgment Prediction (LJP) problem with regard to LLMs coupled with an IR framework, on CAIL (Xiao et al., 2018) dataset.

**Questions For The Authors:**

Question A: Why LJP baselines e.g. Neural Legal Judgment Prediction in English, Chalkidis et al. ACL 2019; Iteratively Questioning and Answering for Interpretable Legal Judgment Prediction, Zhong et al. AAAI 2020; ILDC for CJPE: Indian Legal Documents Corpus for Court Judgment Prediction and Explanation, Malik et al. ACL 2021., are not considered for comparison?
Question B: How are the case facts truncated to 500 tokens? Are they done randomly? Or are the first/last 500 tokens considered?
Question C: Are the results fair w.r.t the under-represented groups? E.g. are females or blacks discriminated by the recommendations of LLMs?
Question D: Are the results acceptable to legal experts? Have legal experts qualitatively/quantitatively analyzed the results? In the midst of
instances where judges are using ChatGPTs for ruling (see https://www.theguardian.com/technology/2023/feb/03/colombia-judge-chatgpt-ruling), apart from accurate prediction, it is absolutely essential to highlight the underlying risks of using such models.
Question E: How about hallucinations by LLMs? Nothing is discussed about the hallucinations of the LLMs.
Question F: What about the explainability of the results?

**Reasons To Accept:**

The paper shows many state-of-the-art LLMs (GPT, Vicuna, ChatGLM, BLOOM) for the legal judgment prediction (LJP) task.

The detailed and comparative performance analysis of LLMs is rigorous.

The interplay of LLMs with IR models esp. identification and analysis of cases where IR outperforms LLMs is detailed.

**Reasons To Reject:**

A baseline of any LJP task should be an existing state-of-the-art LJP method. It would have been interesting to see how LLMs (GPT, BLOOM etc.) presented in this paper live up to much stronger baselines for this task. It is also not clear why the task is framed as an IR task when LJP is usually a supervised problem e.g. as in Neural Legal Judgment Prediction in English, Chalkidis et al. ACL 2019; Iteratively Questioning and Answering for Interpretable Legal Judgment Prediction, Zhong et al. AAAI 2020; ILDC for CJPE: Indian Legal Documents Corpus for Court Judgment Prediction and Explanation, Malik et al. ACL 2021. A comparison of LLMs in this paper with such strong baselines would have been worthy of investigation w.r.t answering questions like "Are GPTs, BLOOM etc. capable of outperforming BERT-based baselines in LJP tasks?"



The paper does highlight the usefulness of LLMs in a legal downstream task. However, it ignores the following key questions:
* Are the results fair w.r.t the under-represented groups? E.g. are females or blacks discriminated by the recommendations of LLMs?
* Are the results acceptable to legal experts? Have legal experts qualitatively/quantitatively analyzed the results? In the midst of instances where judges are using ChatGPTs for ruling (see https://www.theguardian.com/technology/2023/feb/03/colombia-judge-chatgpt-ruling), apart from accurate prediction, it is absolutely essential to highlight the underlying risks of using such models.
* How about hallucinations by LLMs? Nothing is discussed about the hallucinations of the LLMs.
* What about the explainability of the results?

**Reproducibility:**

4: Could mostly reproduce the results, but there may be some variation because of sample variance or minor variations in their interpretation of the protocol or method.

**Reviewer Confidence:**

4: Quite sure. I tried to check the important points carefully. It's unlikely, though conceivable, that I missed something that should affect my ratings.

---

> ### Author Rebuttal · Authors · 2023-08-29
>
> Thanks for your efforts on reviewing our paper. We are grateful for the detailed comments and clear suggestions. As our work is one of the pioneering efforts in LLM legal benchmarks and considering that LLMs are emerging and have been significantly reshaping the field, it is inevitable that our paper is not clear enough to present its motivation and significance.
>
> Your concerns may stem from the dissimilarity between our work and previous Legal NLP work, e.g., judgment prediction. For example, as mentioned in the Reasons to Reject, we do not compare LLMs to supervised baselines and the classification task is tackled with a retrieval system. This paper is not to propose new methods on the LJP tasks; instead, we use the LJP task to measure the legal ability of LLMs. Perhaps our title and the description in the 2nd paragraph that “we design practical baseline solutions based on LLMs” make you confused about the contribution of the paper.
>
> The aim of our work is to propose a benchmark to evaluate the legal ability of LLMs and to give a comprehensive analysis to provide some insights and shield light on future work, especially the retrieval-based LLM applications. Our results, e.g., ablation study of the number of similar cases and the paradox that LLM+IR underperform IR, indicate the importance of the quality of retrieval results, and point out two research directions:
> - how to pretrain LLMs to work with retrieval noise (Wang et al. 2023)
> - how to improve the capability of retrieval systems to provide useful information for LLMs, e.g., searching from structured data like knowledge graph.
>
> As LLM evaluation is subtle and vulnerable to prompting and inference implementations (https://huggingface.co/blog/evaluating-mmlu-leaderboard), we try to focus the paper on LLMs and we cannot include more aspects due to space limits. However, we do have some additional experiments and we will discuss them later.
>
> ### Response to Rejecting Reason “Lack of Baselines”
> We experiment on three baseline models: BERT, XLM-RoBERTa and DeBERTa. We train these models with different sizes of training data: the original CAIL dataset (~100k samples) and the sampled training set (1120 samples), denoted as CAIL_few, described in Section 3.2. The smaller training set is for fair comparison with LLMs, as LLMs are good at few-shot and zero-shot learning and our retrieval system only use the sampled smaller training set. The sampled training set estimates the ability of supervised baselines under low data.
>
> The prediction accuracy is shown below:
> | model | CAIL | CAIL_few|
> | - | - | - |
> | BERT | 89.64 | 68.04 |
> | XLM-RoBERTa | 88.75 | 66.43 |
> | DeBERTa| 88.57 | 30.89 |
>
> Considering the highest accuracy of GPT-4 being 74.46% (multi-choice, 4shot), GPT-4 can outperform supervised baselines in low data scenario. If there is abundant training data, supervised baselines are still better than GPT-4 at 15%.
>
> ### Response to Rejecting Reason “Framing LJP as IR”
> Enhancing LLMs with retrieval systems is an emerging research direction and has been drawing increasing attention (ACL 2023 tutorial: Retrieval-based Language Models and Applications). IR can provide explicit knowledge and help mitigate the hallucination of LLMs (Jiang et al. 2023). Besides, similar demonstrations can also improve in-context learning results (Liu et al. 2022), which are obtained by IR systems. We adopt this common practice and incorporate the IR system to retrieve similar cases. Such similar cases, also known as precedents in legal practice, are valuable for legal judgments.
>
> ### Response to Rejecting Reason about “Fairness, Expert Evaluation, Explainability, and Hallucination”
> These points are critical topics of LLMs and we are also interested in them. However, as they are all hard problems and they are independent research directions, we cannot discuss them in our paper due to the limitation of space and less relevance of our work.
>
> We also notice that there are some work discussing these topics, like Santosh et al. 2022, Wu et al. 2022. We are very pleased to explore these important aspects in our future work.
>
> ### Response to Question A
> Our paper is more to evaluate LLM with LJP than to solve LJP with LLM, so we replace the comparison of baselines with more analysis to give some insights of LLMs. We are pleased to include these comparison in the Appendix.
>
> ### Response to Question B
> We truncate similar cases to preserve the first 500 tokens. We also notice that some other work chooses to truncate beginning tokens. Our truncation setting is based on the heuristics on the CAIL dataset that important case facts usually appear at the beginning followed by some details.
>
> ### Response to Question C, D, E, F
> These comments are discussed in the *Response to Rejecting Reason about “Fairness, Expert Evaluation, Explainability, and Hallucination”*. As they are not relevant to this work, we leave them to be explored in future work.
>
>
> ### References
> - Shall We Pretrain Autoregressive Language Models with Retrieval? A Comprehensive Study. Wang et al. 2023.
> - Active Retrieval Augmented Generation. Jiang et al. 2023.
> - What Makes Good In-Context Examples for GPT-3?. Liu et al. 2022
> - Deconfounding Legal Judgment Prediction for European Court of Human Rights Cases Towards Better Alignment with Experts. Santosh et al. EMNLP 2022
> - Towards Interactivity and Interpretability: A Rationale-based Legal Judgment Prediction Framework. Wu et al. EMNLP 2022

---

### Meta-Review · Area_Chair_Vfci · 2023-09-18

**Recommendation:** 2

**Metareview:**

This paper presents a comprehensive benchmark for evaluating large language models on legal judgment prediction using the CAIL dataset. The work designs prompt-based solutions leveraging LLMs to map case facts to charges. An information retrieval system is incorporated to provide similar cases as context. The experiments analyze model performance under various settings like open vs multiple choice questions and zero-shot vs few-shot learning.

Reviewer 1 highlighted the rigorous comparative analysis of multiple state-of-the-art LLMs as a strength. However, they felt using supervised learning baselines from existing legal judgment prediction literature would make the evaluation more compelling. They also questioned the lack of analysis around critical issues like fairness, expert evaluation, explainability and hallucination risks. In the rebuttal, the authors provide additional experiments with BERT-based models as baselines and acknowledge the importance of the highlighted issues for future work.

Reviewer 2 found the clean experimental design and straightforward performance comparison between models to be merits. A concern raised was whether the charge labeling task evaluated is a practical legal application. The authors clarify in the rebuttal that legal judgment prediction serves as a proxy task requiring multifaceted legal abilities, though other complex applications like case report writing could also be considered. The reviewer also sought more clarity on the implications of the "paradox" finding. The authors explain this points to the importance of retrieval quality and future research directions.

Reviewer 3 recognized the comprehensive analysis of prompting strategies as a strength. They suggested comparing results to general domain prompting and highlighting legal-specific challenges. The rebuttal provides examples of legal reasoning difficulties like charges with overlapping facts and penalties. The reviewer also asked about tokenizer differences which the authors analyze by correlating sequence length and performance.

I agree with the reviewers that the paper is beset with  some of these limitations:
- Lack of comparison to supervised baselines on LJP (Review 1). The authors provide additional experiments with BERT-based models in the rebuttal.
- Unclear implications/applications of the "paradox" finding (Review 2). The authors clarify the takeaways for future work.
- Overlap with general domain prompting effects without highlighting legal-specific challenges (Review 3). The rebuttal provides more legal case analysis.


Summarily, the reviewers recognize this work provides one of the first comprehensive LLM benchmarks tailored for the legal domain, which is a novel contribution. While comparisons to supervised learning baselines are lacking, the authors provide additional experiments in the rebuttal indicating LLMs can outperform BERT-based models under low resource settings. They also acknowledge the importance of analyzing issues like fairness and plan to investigate them in future work.

The remarks about practical applications and implications of findings help highlight areas where the paper could be strengthened, but the core utility of extensive LLM analysis for an under-explored domain remains. The authors have made efforts in their responses to provide more legal case insights and clarify the takeaways.

Considering the authors' effort in addressing some of these issues in the rebuttal, I would encourage them to carry out thorough revisions to enrich the legal analysis, highlight unique challenges, and crystallize the implications of findings that could make this a stronger contribution.

---

### Decision · Program_Chairs · 2023-10-07

**Decision:**

Accept-Findings

**Comment:**

This paper presents a comprehensive benchmark for evaluating large language models on legal judgment prediction using the CAIL dataset. The work designs prompt-based solutions leveraging LLMs to map case facts to charges. An information retrieval system is incorporated to provide similar cases as context. The experiments analyze model performance under various settings like open vs multiple choice questions and zero-shot vs few-shot learning.

Reviewer 1 highlighted the rigorous comparative analysis of multiple state-of-the-art LLMs as a strength. However, they felt using supervised learning baselines from existing legal judgment prediction literature would make the evaluation more compelling. They also questioned the lack of analysis around critical issues like fairness, expert evaluation, explainability and hallucination risks. In the rebuttal, the authors provide additional experiments with BERT-based models as baselines and acknowledge the importance of the highlighted issues for future work.

Reviewer 2 found the clean experimental design and straightforward performance comparison between models to be merits. A concern raised was whether the charge labeling task evaluated is a practical legal application. The authors clarify in the rebuttal that legal judgment prediction serves as a proxy task requiring multifaceted legal abilities, though other complex applications like case report writing could also be considered. The reviewer also sought more clarity on the implications of the "paradox" finding. The authors explain this points to the importance of retrieval quality and future research directions.

Reviewer 3 recognized the comprehensive analysis of prompting strategies as a strength. They suggested comparing results to general domain prompting and highlighting legal-specific challenges. The rebuttal provides examples of legal reasoning difficulties like charges with overlapping facts and penalties. The reviewer also asked about tokenizer differences which the authors analyze by correlating sequence length and performance.

I agree with the reviewers that the paper is beset with  some of these limitations:
- Lack of comparison to supervised baselines on LJP (Review 1). The authors provide additional experiments with BERT-based models in the rebuttal.
- Unclear implications/applications of the "paradox" finding (Review 2). The authors clarify the takeaways for future work.
- Overlap with general domain prompting effects without highlighting legal-specific challenges (Review 3). The rebuttal provides more legal case analysis.


Summarily, the reviewers recognize this work provides one of the first comprehensive LLM benchmarks tailored for the legal domain, which is a novel contribution. While comparisons to supervised learning baselines are lacking, the authors provide additional experiments in the rebuttal indicating LLMs can outperform BERT-based models under low resource settings. They also acknowledge the importance of analyzing issues like fairness and plan to investigate them in future work.

The remarks about practical applications and implications of findings help highlight areas where the paper could be strengthened, but the core utility of extensive LLM analysis for an under-explored domain remains. The authors have made efforts in their responses to provide more legal case insights and clarify the takeaways.

Considering the authors' effort in addressing some of these issues in the rebuttal, I would encourage them to carry out thorough revisions to enrich the legal analysis, highlight unique challenges, and crystallize the implications of findings that could make this a stronger contribution.